# Motion and teleportation of polar bubbles in low-dimensional ferroelectrics

S. Prokhorenko [1] ✉, Y. Nahas [1], V. Govinden[2], Q. Zhang[2,3] ✉, N. Valanoor [2] & L. Bellaiche [1]

Electric bubbles are sub-10nm spherical vortices of electric dipoles that can spontaneously form in ultra-thin ferroelectrics. While the static properties of electric bubbles are well established, little to nothing is known about the dynamics of these particle-like structures. Here, we reveal pathways to realizing both the spontaneous and controlled dynamics of electric bubbles in ultra-thin $Pb(Zr_{0.4}Ti_{0.6})O_3$ films. In low screening conditions, we find that electric bubbles exhibit thermally-driven chaotic motion giving rise to a liquid-like state. In the high screening regime, we show that bubbles remain static but can be continuously displaced by a local electric field. Additionally, we predict and experimentally demonstrate the possibility of bubble teleportation - a process wherein a bubble is transferred to a new location via a single electric field pulse of a PFM tip. Finally, we attribute the discovered phenomena to the hierarchical structure of the energy landscape.

Dipolar topological patterns in ferroelectrics hold extraordinary technological potential for information storage and processing due to their nanometer length scale, stability, and electric field sensitivity. Of particular interest are the so-called polar or electric bubbles, theoretically predicted[1,2] to spontaneously form in $Pb(Zr,Ti)O_3$ (PZT) thin films[3]. Two closely resembling varieties of electric bubbles (achiral and chiral) were also recently observed at room temperature in $Pb(Zr,Ti)O_3/SrTiO_3$ heterostructures[4] and $PbTiO_3/SrTiO_3$ superlattices[5].

Here, we focus on achiral bubbles typical for $Pb(Zr_{0.4}Ti_{0.6})O_3$-based systems. Structurally, such a bubble can be described as a three-dimensional vortex composed of local electric dipoles (Fig. 1a). It consists of a core axis oriented perpendicular to the films' surface wherein the local dipoles are pointing along up (or down) out-of-plane direction. The core is belted by a toroidal vortex, which can be visualized as a dipolar vortex tube deformed and glued into a doughnut shape[3,6]. The whole structure is confined within a spherical boundary on which the dipoles follow the meridian lines in the direction opposite to the core polarization. The latter allows for a seamless embedding of the bubble in a homogeneously down (or up) polarized matrix domain. Thereby, the structure of a bubble closely resembles the distribution of flux lines in the so-called Hill's spherical vortex[7]—a model soliton meant to approximate the structure of vortex rings in fluids and gases. Alternatively, the polar structure of a bubble can be compared with the distribution of electric field lines in the so-called anapoles[8].

Topologically, electric bubbles share similarities with magnetic skyrmions[9]. For example, a constant latitude cross-section of a bubble produces a dipolar skyrmion texture characterized by an integer Skyrmion number[5,10]. Due to this fact, electric bubbles are also termed polar skyrmions[10]. To illustrate this point, we show in Fig. 1b, c a simulated structure of a hexagonal bubble lattice (see Methods) in a five u.c. thick $Pb(Ti_{0.4}Zr_{0.6})O_3$ film. More specifically, Fig. 1b presents the distribution of local electric dipoles within a plane located one u.c. below the top interface, while Fig. 1c shows the same distribution within a plane one u.c. above the bottom interface. These latitudinal cross-sections allow us to clearly see the Néel skyrmion textures produced by each bubble. Further details on the structure of electric bubbles can be found in recent review articles[3,11].

As of today, the static properties of electric bubbles are well understood. Particularly, we know that standalone bubbles can be deterministically written and erased[12] or spontaneously form bubble arrays and lattices under external bias[4,10,13,14]. Notably, the latter phenomenon is remarkably similar to the formation of magnetic skyrmion

---

[1]Physics Department and Institute for Nanoscience and Engineering, University of Arkansas, Fayetteville, AR 72701, USA. [2]School of Materials Science and Engineering, The University of New South Wales, Sydney, NSW 2052, Australia. [3]CSIRO Manufacturing, Lindfield, NSW 2070, Australia. ✉e-mail: sprokhor@uark.edu; peggy.zhang@csiro.au

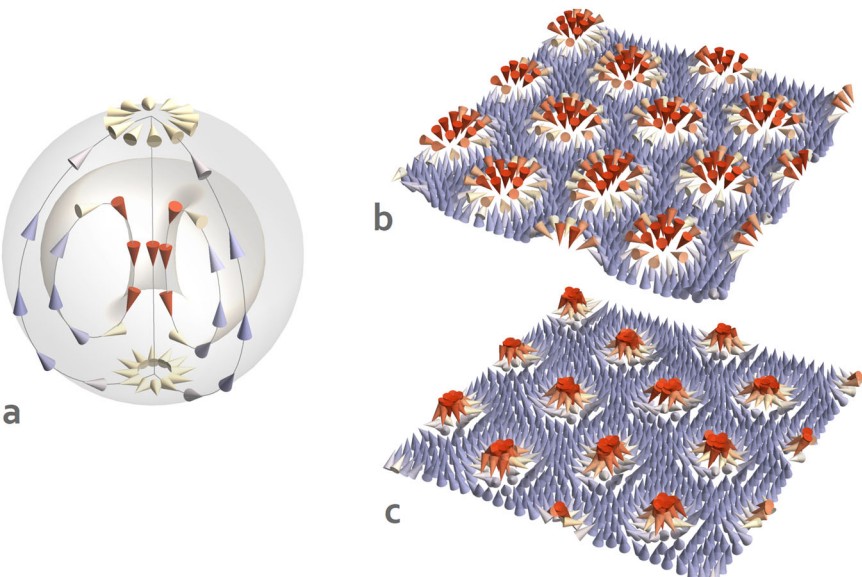

**Fig. 1 | Dipolar structure of electric bubbles. a** Schematic structure of an electric bubble. The arrows represent electric dipoles colored according to their out-of-plane Cartesian component. The downwards-pointing dipoles (red arrows) correspond to the core of the bubble. The light-gray surfaces correspond to the inner toroidal vortex (light gray torus) surrounding the core and the outer boundary of the bubble (light gray sphere). The electric bubble is confined between the top and bottom interfaces of the PZT film (not shown) so that the core is polarized along the out-of-plane $z$-axis. Electric bubbles most often form partially ordered arrays or hexagonal lattices. **b, c** The simulated planar cross-sections of a hexagonal bubble lattice one u.c. above (**b**) and below (**c**) the middle plane of the film. Within such cross-sections, the bubble gives rise to a Néel skyrmion pattern (red arrows). The skyrmions above and below the equatorial plane feature the same polarity and a mutually opposite sense of Néel rotations. In panels **b** and **c**, local dipoles are represented by arrows colored according to the dipole's out-of-plane component.

patterns in thin films of chiral magnets[9]. At the same time, the dynamics of electric bubbles still remain unexplored. For example, it is not clear whether electric bubbles can spontaneously move like their magnetic siblings in the so-called skyrmion liquids[15]? Or, can one rather induce a deterministic motion of standalone bubbles akin to the current-driven displacement of magnetic skyrmions[16]? These unanswered questions are further supported by the recent reports of both the spontaneous[13] and induced[17] motion of electric dislocations or merons[10]. Such dipolar structures are essentially half-bubbles attached to the end of a polar vortex tube[6,18]. While having fundamental importance, the particle-like dynamics bubble dynamics could open doors to new technological applications. Indeed, controlled bubble motion could enable dynamically re-configurable electronic circuits[19], while the spontaneous motion of electric bubbles can be employed in stochastic computing[20]. Finally, the discovery of bubble dynamics could also entail new physical phenomena. For example, having in mind the coupling of electric bubbles to electrons[15], one might wonder whether dynamical bubble states could reveal novel electron phases. Motivated by these prospects, we here explore ways to realize both spontaneous and controlled motion of electric bubbles.

## Results

First, we consider the possibilities of spontaneous motion of electric bubbles at room temperature. For this, we note that the spontaneous motion of magnetic skyrmions was observed in the vicinity of the skyrmion lattice melting triggered by an external magnetic field[15] when the skyrmion stability is significantly weakened. Thus, it appears promising to consider bubble states in the vicinity of field-induced transition regions.

### Stability of bubble states with varying bias and screening

In order to tune the stability of bubble states[6], we use a combination of two external parameters—the constant out-of-plane electric field $E_z$ and the screening of interfacial bound charges $\beta$. Both $E_z$ and $\beta$ were previously shown to be of utmost importance for the stability of

bubble states[1,2,4,10,21]. Furthermore, both parameters can be tuned in experiments. For instance, changing the screening can be achieved by sandwiching the PZT film with $SrTiO_3$ slabs of varying thickness[21]. Here, in order to get a complete picture, we numerically compute the full ($E_z, \beta$) phase diagram at room temperature. To obtain this phase diagram, we use ab initio-based effective Hamiltonian simulations[22]. In these simulations, we employ a $32 \times 32 \times 5$ supercell to mimic a 2nm thick PZT film with in-plane periodic boundary conditions. The interfacial screening $\beta$ is introduced within an atomistic approach to depolarizing field calculations in ultra-thin ferroelectrics[23]. Physically, $\beta$ can be seen as the fraction of the bound charge screened at the surfaces of the two-dimensional structure, with $\beta = 0$ corresponding to the ideal open circuit and $\beta = 1$ to the ideal short circuit electrical boundary conditions. Finally, the film is assumed to be under a −2% epitaxial strain to mimic epitaxial growth on a $SrTiO_3$ substrate. Further technical details are described in the Methods section.

The variation of the bubble density with $E_z$ and $\beta$ obtained from Monte Carlo relaxation is presented in Fig. 2a. The corresponding phase diagram is shown in Fig. 2b. As it can be seen, at low screening and bias magnitudes, the system features a nanostripe domain structure (St). The stripes progressively break upon increasing either $E_z$ or $\beta$, which results in a hexagonal bubble lattice[14] (bub). Finally, at high bias magnitudes and good screening conditions, our simulations predict a homogeneously polarized state (FE). Hereon, we will denote the bias values corresponding to St-bub and bub-FE transitions as $E_{bub}$ and $E_{FE}$, respectively.

On the one hand, these results are fully in line with previously reported data. Indeed, the described sequence of stripe-bubble-monodomain ((St-bub-FE)) transitions is well documented[1,2,6,10]. On the other hand, the calculated phase diagram reveals two presently unknown special points, which, as we will show below, are manifestations of a new physical phenomenon.

The first special point is marked by a blue circle in Fig. 2b and corresponds to a tri-critical point on the bub-FE line. It occurs at $\beta_\star \approx 0.86$. Consequently, the bub-FE transition is continuous for $\beta < \beta_\star$

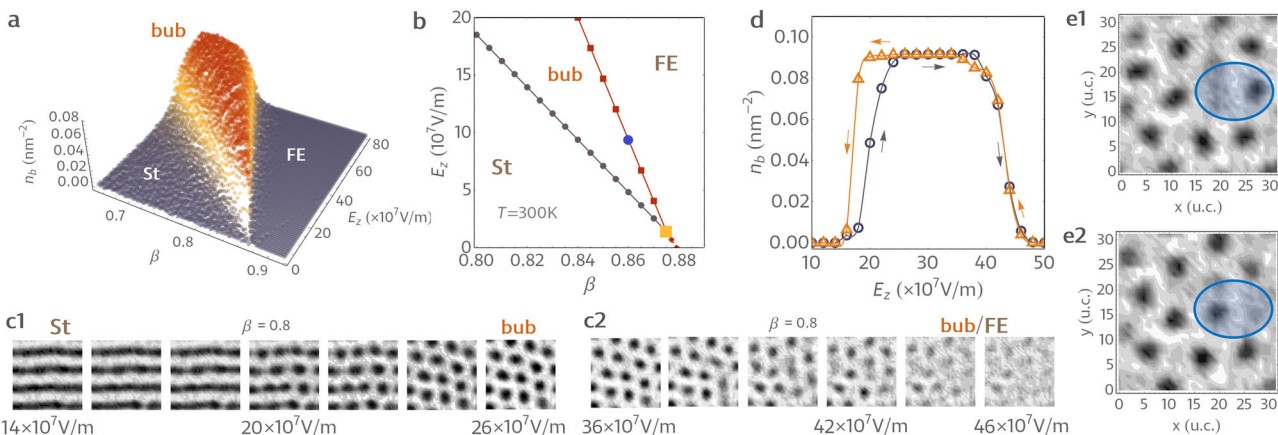

**Fig. 2 | Room temperature phase diagram and dynamical topological excitations. a** Variation of the bubble density $n_b$ with external field magnitude $E_z$ and screening strength $\beta$. The region of high $n_b$ values corresponds to the hexagonal bubble lattice state (denoted as "bub") separating the vortex tube or stripe phase "St" from the homogeneously polarized state "FE". For both the St and FE phases, the bubble domain density $n_b$ is zero. **b** The computed bias field-screening phase diagram at $T = 300$ K. The blue circle marks a tri-critical point. The yellow square indicates the crossing of transition lines. **c1, c2** The distribution of the out-of-plane component of electric dipoles within the middle plane of the $32 \times 32 \times 5$ supercell (12.8 nm × 12.8 mm × 2 nm) at the different field values. The bright and dark regions correspond to domains of opposite (up/down) polarity. The dipolar structure snapshots in panels **c1** and **c2** are obtained for $E_z$ values range of $14–26 \times 10^7$ V/m and $36–46 \times 10^7$ V/m with steps of $2 \times 10^7$ V/m, respectively. **d** The field hysteresis of bubble domain density obtained from molecular dynamics simulations at $\beta = 0.8$ and $T = 300$ K. The increasing (decreasing) field branches are plotted with dark gray circles (yellow triangles). Yellow and gray lines are guides for the eye. **e** Maps of the running average of polarization within the middle plane of the supercell obtained from molecular dynamics simulation at $\beta = 0.8$, $T = 300$ K, and $E = 40 \times 10^7$ V/m. Panels **e1** and **e2** correspond to times $t = 8$ ps and $t = 9$ ps, respectively. The averages are performed over 20 configurations within the running window of 0.5 ps. The blue circle highlights a region of a dynamic bubble displacement characteristic of the second-order transition at $\beta > \beta_\star$.

and discontinuous for $\beta > \beta_\star$. Notably, the position $\beta_\star$ of the tri-critical point is temperature dependent. For instance, at 10 K, we find that the bub-FE transition is discontinuous for all of the considered $\beta$ values. Therefore, $\beta_\star$ moves toward lower screening conditions with decreasing temperature. The second special point corresponds to the intersection of the St-bub and bub-FE transition lines (marked by a yellow square in Fig. 2b). As the screening is enhanced, the two critical fields linearly decrease but also get closer together until coinciding at $\beta \approx 0.875$. In the case when screening is above this threshold point, our simulations predict a single $E_z$-triggered first-order phase transition line separating stripe domains from a homogeneously polarized state.

In fact, the intersection point is related to the change of transition behavior at $\beta_\star$. At the first-order bub-FE transition, the polarization experiences a jump. Similarly, the total area $R$ of the film switched along the field abruptly changes from below 100% to 100% at $E_{\mathrm{FE}}$. Such discontinuity is all the more pronounced with increasing $\beta$. In other words, the onset of the FE state is pushed to lower $R$ values with increasing $\beta$. At the same time, the bubble lattice is geometrically constrained[10] to occur for 69% < $R$ < 100% (Supplementary Fig. 1). As a result, the intersection point occurs when the FE state onsets at exactly $R = 69\%$.

The key finding is thus the much unexpected tri-critical point. As a matter of fact, the possibility of a 2nd order transition from the bubble lattice to a homogeneously polarized state appears somewhat puzzling. Indeed, previous studies have reported that bub-FE transition has a first-order character[10] with both the density of bubbles and polarization showing a field hysteresis[10,13] in the vicinity of $E_{\mathrm{FE}}$. Such behavior was naturally expected vis à vis the binodal nature of the $E_{\mathrm{FE}}$ line[10], but also due to the topological stability of electric bubbles. Particularly, the latter was shown to yield multiple meta-stable minima in the vicinity of $E_{\mathrm{FE}}$ corresponding to different realizations of low-density bubble arrays[10]. Such static metastable states are signatures of phase coexistence known to occur at the first-order discontinuous transitions.

### Spontaneous bubble motion at the second-order transition
To clarify the origin of the tri-critical point, we thus inquire into the microscopic mechanism underlying the continuous bub-FE

transformation. For this, we first inspect the evolution of the polar pattern with $E_z$ for $\beta < \beta_\star$. Figure 2c1–c2 shows the calculated distribution of the out-of-plane polarization within the middle plane of the film at different values of the applied bias. At first sight, we observe the previously reported sequence of the St-bub-FE transformations[10] mediated by the gradual breaking of the polar stripes[6]. However, we can also note some distinctive features of the dilute bubble arrays in the vicinity of the bub-FE transition (Fig. 2c2). For instance, we can note that some bubbles in Fig. 2c2 have "fuzzy" boundaries (e.g., at $E_z = 40 – 42 \times 10^7$ V/m in Fig. 2c2). Moreover, the polarization contrast for dilute bubble arrays is much lower than that of the bubble lattice (e.g., $E_z = 42 \times 10^7$ V/m vs. $E_z = 36 \times 10^7$ V/m in Fig. 2c2). Such features are not seen in the previously reported dilute bubble arrays at the discontinuous transition regime[6,10,13].

As a next step, we performed molecular dynamics simulations (see Methods) of a hysteresis field cycle for $\beta = 0.8$ at 300 K. For this, we have used the same effective Hamiltonian model, supercell geometry, and the external parameter values as in the Monte Carlo relaxations used to obtain the phase diagram (Fig. 2a, b). The zero field molecular dynamics was initiated from the nanostripe configuration obtained from the previously described Monte Carlo relaxation. Then, we progressively increased the applied field $E_z$ from zero to $50 \times 10^7$ V/m with increments of $5 \times 10^7$ V/m. At each field value, we have performed 150,000 integration steps (total trajectory time of 75 ps), with the last 50,000 steps (25 ps) used to obtain statistics. Finally, the decreasing field branch of the cycle was performed in a similar fashion.

The calculated hysteresis of the average density of bubbles $n_b$ vs. applied field is shown in Fig. 2d, where the increasing and decreasing field branches are plotted with gray and yellow symbols, respectively. Here, one can immediately note that $n_b$ exhibits a hysteresis at the transition from nanostripes to bubbles. This feature is due to a relatively short molecular dynamics relaxation time (25 ps) that we used to accentuate possible hysteretic features. Indeed, our Monte Carlo simulations do not show any hysteresis of $n_b$ at $E_{\mathrm{bub}}$ for $\beta = 0.8$. Therefore, the described discrepancy between the two branches is rather indicative of slow relaxation dynamics. At the same time, despite a fast rate of field change, the density variation in the vicinity of

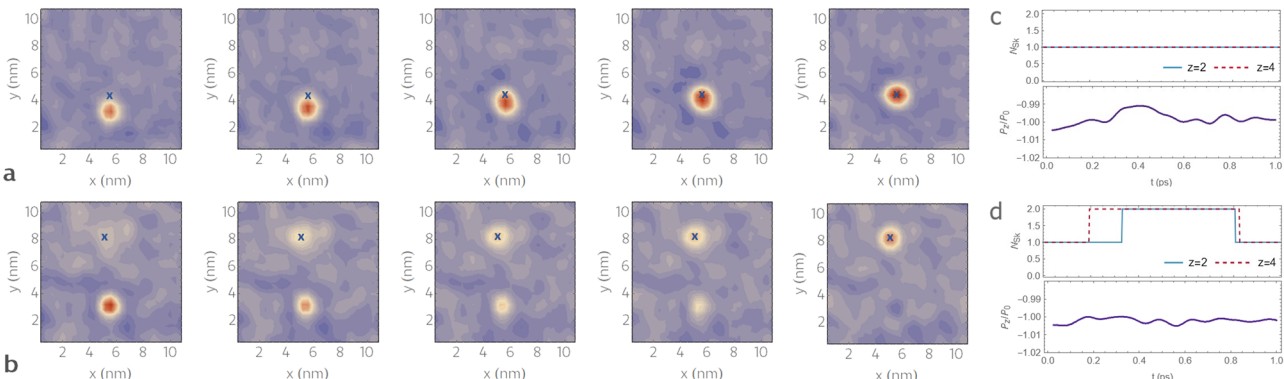

**Fig. 3 | Electric-field-induced motion of polar bubbles. a** Simulation of short-range displacement of the bubble induced by the PFM tip located 2 nm away from the bubble center (position of the tip indicated with a blue cross). The driving field profile with a maximum local value of $21 \times 10^7$ V/m is assumed. As it can be seen, the bubble displaces from its initial position (left-most image) towards the point right beneath the PFM tip (right-most image) through a stretch-like motion (middle image). **b** Time evolution of the structure when PFM tip (blue cross) is applied 5.5 nm away from the bubble center. Here, the local maximum of the field profile is of $110 \times 10^7$ V/m. In this case, the local electric field creates a bubble domain beneath the tip while simultaneously deleting the bubble domain at its initial position. In panels, **a** and **b**, the left to right sub-panels show the distribution of the $z$ component of polarization at times $t = 0$, 0.6 ps, 0.75 ps, 0.8 ps, and 1 ps, respectively. Blue to red colors correspond to negative to positive polarization values. **c**, **d** Evolution of the Skyrmion number within the $z = 2$ and $z = 4$ planes and the normalized out-of-plane polarization. Panel (**c**) and (**d**) correspond to the continuous motion and teleportation, respectively. All presented simulation results were performed at 10 K and a background bias field $E_z$ of $61 \times 10^7$ V/m.

$E_{FE}$ does not show any pronounced hysteretic behavior. This observation confirms that the transformation from a bubble lattice to a homogeneously polarized state at $\beta = 0.8$ is indeed a continuous second-order transition. To explore the corresponding critical fluctuations, we have visualized the molecular dynamics evolution of the dipolar structure at $E_z = 40 \times 10^7$ V/m. Two typical structural snapshots are shown in Fig. 2e1–e2. Both panels show the distribution of the out-of-plane components of dipoles in the middle plane of the film. In both cases, we see a bubble lattice structure with some vacancies but also immediately notice that the position of electric bubbles is different in Fig. 2e1 and Fig. 2e2 (e.g., the blue region in Fig. 2e1–e2. As a matter of fact, an animation of the structural evolution (Supplementary Movie 1) clearly indicates that the bubble array is constantly changing in time as bubbles chaotically displace. Moreover, we notice that, from time to time, bubbles seem to "hop" from one position to another. Such a dynamic process can be described as a progressive fading of a bubble accompanied by a synchronous emergence of another bubble in a nearby location. This process is visually remindful of teleportation.

Therefore, we can conclude that critical fluctuations under low screening conditions ($\beta < \beta_\star$) give rise to a dynamical phase of electric bubbles, which is, in some aspects, very similar to the magnetic skyrmion liquid[15].

Notably, the spontaneous bubble dynamics at the continuous bub-FE transition explain the fuzzy structural contrast in Fig. 2c2. Indeed, a chaotically moving bubble leaves traces in the structure averaged over multiple Monte Carlo sweeps and appears to occupy several nearby locations at the same time. For instance, a single bubble moving back and forth between two positions gives rise to peanut-shaped gray features in Fig. 2c2 (e.g., $E_z = 38 \times 10^7$ V/m).

**Field-driven motion at the first-order transition**

While the continuous transition entails spontaneous bubble dynamics, the vicinity of the first-order bub-FE topological transformation appears to be promising for the controlled bubble motion. Indeed, as previously shown[10], tuning the bias in the vicinity of the first-order transition line allows low-density arrays of static bubbles to be obtained. In other words, one readily obtains access to well-isolated bubbles that would retain their position in the absence of external stimuli.

The most natural possibility to consider is the motion induced by a *local* electric field oriented along the dipole moment of the bubble. If such a field is applied close to the bubble, we naturally expect the bubble to drift toward the point of the highest field magnitude. To test this idea, we have performed molecular dynamics simulations with a local driving field profile approximating that of a Piezoresponse force microscopy (PFM) probe[24]. The global electric field bias was chosen so as to ensure a single electric bubble within the supercell. The simulations are performed for $\beta = 0.8$ but at the temperature of 10 K. The low temperature is chosen to separate the effects of thermal fluctuations from that of the field perturbation. Under these conditions, the bub-FE transition is of the first order for all of the considered $\beta$ values. The corresponding phase diagram obtained from Monte Carlo simulations is shown in Supplementary Fig. 2.

The results presented in Fig. 3 reveal two distinct effects provoked by the considered electric field perturbation, depending on its magnitude and the distance of the tip from the center of the bubble domain. Namely, for a tip located up to 3 nm away from the original bubble location and for a maximum electric field magnitude below the coercive field value, we find that the driving field entails a continuous displacement of the bubble (see Fig. 3a). During such a displacement, the bubble first elongates towards the tip position before contracting once the polarization within the area beneath the tip is reversed. During this process, the overall Skyrmion charge is conserved (see Fig. 3c). Performing the linear fits of the evolution of the bubble position with time (Supplementary Fig. 3), we find that the bubble velocity during such motion increases with increasing driving field magnitude and is of the order of 2 nm/ps or 2000 m/s (Supplementary Fig. 4).

In the case where the distance between the bubble center and the tip is larger than 3 nm, our simulations also predict a bubble creation/annihilation process as seen in Fig. 3b. Under these conditions, a maximum driving field magnitude is required to exceed the coercive field. Specifically, we find that a new bubble domain is progressively created beneath the simulated PFM tip while the bubble at the initial position progressively disappears. This process is similar to spontaneous jumps observed at the continuous bub-FE transition (Fig. 2e). Such *transfer* of the switched area is characterized by a synchronous growth and erasure of the new and original bubbles, respectively, as if the polar skyrmion was being teleported towards the PFM tip. Here, by teleportation, we mean the transfer, without traversing of the physical space, of the state and energy associated with the inhomogeneous dipolar order. During this process, the overall polarization shows lower

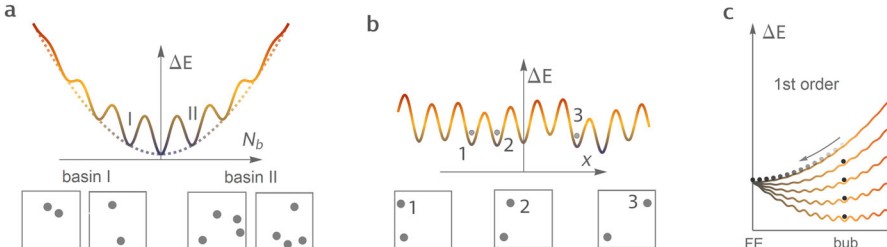
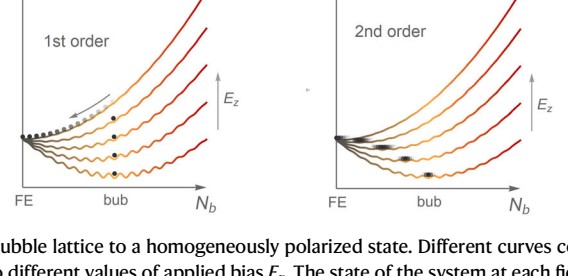

**Fig. 4 | Hierarchical energy landscape of bubble states. a** Dependence of the internal energy $\Delta E$ on the number of bubbles $N_b$. Local minima (basins) correspond to sets of microscopic states with a fixed number of bubbles $N_b$. Examples of microscopic states located in basins with $N_b = 2$ (basin I) and $N_b = 4$ (basin II) are shown below the $\Delta E$ graph. Here, the bubbles are represented by gray circles. In the vicinity of the global minimum, the lowest energies of the basins follow a parabolic envelope function shown as a dotted line. Each basin is further fragmented into local sub-minima corresponding to different spatial distribution $X$ of bubbles. The internal structure of each basin has the shape of a Peierls-Nabarro potential schematically shown for $N_b = 2$ basins in panel (**b**). Here, the local minima 1–3 correspond to different positions $x$ of one of the bubbles, as shown in the subpanels below the graph. **c** Mechanism of the discontinuous first-order phase transition from a bubble lattice to a homogeneously polarized state. Different curves correspond to different values of applied bias $E_z$. The state of the system at each field is represented by a black dot. As $E_z$ increases, the internal energy profile is tilted while the global minimum shifts to lower $N_b$ values. However, due to the barriers between neighboring $N_b$ basins, the system remains stuck in a metastable bubble lattice phase (bub). An abrupt transition to the stable homogeneous state (FE) occurs only when the field-induced inclination of $\Delta E$ flattens out the barriers. **d** For lower barrier heights, the thermal fluctuations allow the system to jump from one basin to another. The corresponding transitions are schematically shown as fuzzy black dots. For each field magnitude, the system is thus able to find its global energy minimum. In this case, the field-induced transition from the bubble lattice to a homogeneous state has a continuous second-order character.

variation than during the continuous motion, while the overall Skyrmion charge conservation within the films' planes is violated (Fig. 3d).

**Origins of tri-critical behavior and underlying bubble dynamics**
Thus far, we have demonstrated that the change of the bub-FE transition character can be ascribed to the change in the microscopic behavior of electric bubbles. Namely, the continuous second-order transition is characterized by critical fluctuations that lead to chaotically moving bubbles. In contrast, the first-order regime gives rise to meta-stable states wherein the bubbles are static but can be displaced by an external perturbation.

To rationalize these findings, we propose to look at different realizations of bubble patterns as different microscopic states of the PZT film. Each of these microscopic states can be uniquely described by the overall number of bubbles $N_b$ (equivalently, the bubble density $n_b$) and the set of positions $X = \{x_1, x_2, \ldots, x_{N_b}\}$ indicating the location of each bubble.

States with different numbers of bubbles $N_b$ have different internal energies $\Delta E$. Indeed, adding or removing one bubble slightly changes the overall out-of-plane polarization $P_z$ of the film. This, in turn, changes the Landau's potential[6]

$$F \approx P_z^2 / \varepsilon(\beta) - E_z P_z, \qquad (1)$$

where $\varepsilon$ denotes the screening-dependent zero-field susceptibility. Additionally, the states with different $N_b$ are separated by energy barriers. Such barriers are related to the energy cost of dipole switching during the bubble creation/removal. Based on these two arguments, we can schematically represent $\Delta E$ as a parabolic curve with multiple local minima (basins), as shown in Fig. 4a. In this graph; we do not consider the dependence of $\Delta E$ on $X$. Therefore, the lowest point of each basin corresponds to set of states with a fixed integer number of bubbles $N_b$ rather than a single microscopic state. Examples of microscopic states within such sets for $N_b = 2$ and $N_b = 4$ basins are shown below the $\delta E$ graph in Fig. 4a.

We now turn to the dependence of the energy on the spatial distribution $X$ of bubbles. Such dependence determines the internal structure of each $N_b$ basin (Fig. 4a) and is schematically shown in Fig. 4b. In fact, each basin in Fig. 4a consists of multiple sub-minima (Fig. 4b) that correspond to various placements $X$ of $N_b$ bubbles. The depths of this sub-minima slightly change with $X$ as a result of two factors—(1) the change in a local potential experienced by each bubble

and (2) the interaction between the bubbles. The former factor is due to different atomic environments (Zr/Ti distribution) in the PZT alloy and, possibly, structural defects such as oxygen vacancies. The second factor comes from the change of distances between bubbles upon changing $X$. Finally, the barriers between sub-minima in Fig. 4b are determined by the energy needed to displace a single bubble from one position to another. Equivalently, the internal structure of each basin is analogous to the Pierls–Nabarro potential associated with the motion of, e.g., dislocations[25] or ferroelectric domain walls[26,27].

Overall, taking into consideration the dependence of energy on $N_b$ and $X$, we conclude that the energy landscape of bubble states has a hierarchical structure with multiple basins (Fig. 4a), each fragmented into an exponentially large number of sub-minima (Fig. 4b). Such structure explains our prediction of the bub-FE transition order crossover as well as the underlying change in the behavior of bubbles.

Indeed, both the position of the lowest energy basin and the barrier heights in Fig. 4a are determined by the applied bias $E_z$ and the screening conditions $\beta$ (Eq. (1)). The effect of these variables on the energy profile can be understood by looking at the envelope function which follows the position of the basins' minima (dotted line in Fig. 4a). Due to the linear coupling of $E_z$ and $N_b$ (Eq. (1)), the increasing bias shifts the global minimum of the curve to lower $N_b$ values and changes the slope at $N_b = 0$. In contrast, the change of screening conditions modifies the curvature of the $\Delta E$ envelope ($1/\varepsilon$ coefficient in Eq. (1)). Specifically, the energy profile becomes steeper when screening is reduced. As a result, the barriers between neighboring basins are effectively reduced at higher $E_z$ and lower $\beta$ values.

The described changes of the barrier heights allow for a crossover from the first to the second order bub-FE transition with changing $\beta$. Namely, under good screening conditions ($\beta > \beta_*$), the barriers are high enough to prevent thermal fluctuation-induced transitions between the nearby $N_b$ basins. Moreover, the intra-basin transitions between sub-minima (Fig. 4b) are not possible. As a result, the static bubble lattice can remain in a metastable state even at high $E_z$ values. In such a scenario, the transition from the bubble to the homogeneously polarized state is discontinuous, as schematically shown in Fig. 4c. Moreover, in the vicinity of the critical field, one observes multiple meta-stable minima corresponding to states with different bubble densities and different spatial distributions of bubbles. This regime allows for deterministic writing and the erasure of individual static bubbles[12].

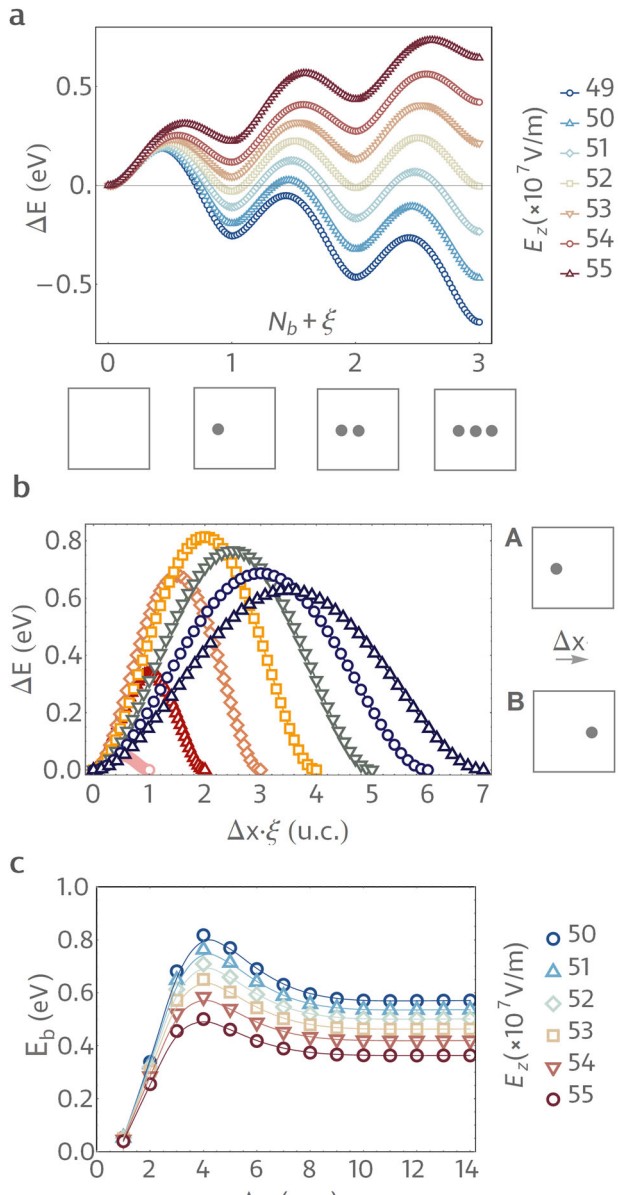

**Fig. 5 | Energy barriers from NEB simulations. a** Calculated total internal energy profile along the NEB paths connecting states with different numbers of bubble domains $N_b$. Differently colored curves correspond to different values of the applied bias field $E_z$. For each $E_z$ value, $\Delta E$ corresponds to the difference between energies of the state on the transition path and the homogeneously polarized state ($N_b = 0$). Schematic representations of states corresponding to different $N_b$ values are shown in subpanels below the graph. **b** The total internal energy variation along the transition paths between the single bubble states where initial (state A) and final bubble locations (state B) are separated by $\Delta x$ unit cells along [001] p.c. direction. All curves are obtained at the bias field $E_z = 50 \times 10^7$ V/m. **c** Variation of the barrier for bubble displacement with the distance $\Delta x$ at different bias magnitudes. In panels, **a** and **b**, $\xi$ denotes the normalized distance along the NEB path with $\xi = 0$ and $\xi = 1$ corresponding to initial and final states, respectively.

energy is also sufficient to overcome the barriers between different sub-minima within a single $N_b$ basin. As a result, thermal fluctuations allow for the spontaneous motion of bubbles within the critical region of $E_z$ values.

To confirm the described mechanism and estimate the main characteristics of the energy landscape, we have performed nudged elastic band (NEB) simulations within the effective Hamiltonian framework[28]. Since the vast configurational space with more than 30,000 degrees of freedom can pose problems with convergence, we have chosen to focus on the limit of low-density bubble states with one to three bubbles in the supercell ($n_b = 0.006$ nm$^{-2}$ to $0.018$ nm$^{-2}$). Moreover, to simplify the analysis, we eliminate the effects of chemical disorder in PZT alloys by using the virtual crystal alloy (VCA) effective Hamiltonian[22]. Such approximation is justified by the recent PFM experiments[12].

The computed energy variation along the NEB paths connecting states with different numbers of bubbles is shown in Fig. 5a. These calculations are performed for different $E_z$ values in the vicinity of $E_{FE}$. As one can see, the slope of the envelope function at $N_b$ is indeed governed by $E_z$. The negative $\Delta E$ slope for $E_z < 52 \times 10^7$ V/m increases with the bias magnitude and becomes positive for $E_z > 52 \times 10^7$ V/m. At $E_z \approx 52 \times 10^7$, the derivative of the envelope function at $N_b = 0$ almost vanishes so that the polar state with low bubble densities becomes quasi-degenerate. The slope of the envelope also determines the barriers for the transition between neighboring $N_b$ basins. For example, the barrier associated with erasing one bubble from a two-bubble ($N_b = 2$) state is of the order of 0.5 eV at $E_z = 49 \times 10^7$ V/m and is reduced to about 0.2 eV at $E_z = 55 \times 10^7$ V/m.

Having estimated the magnitude of energy barriers between basins, we now explore the intra-basin structure. Figure 5b shows the calculated energy profiles associated with bubble displacements for $N_b = 1$ at $E_z = 50 \times 10^7$ V/m and $\beta = 0.8$. In these calculations, the initial state contains one arbitrarily placed bubble (state A in Fig. 5b). In the final state (state B in Fig. 5b), the bubble is displaced by $\Delta x$ unit cells along the [100]$_{p.c.}$ direction. The initial approximation of the transition path corresponds to a linear superposition of the initial and final states.

The total energy variations for different displacements $\Delta x$ are plotted with different colors and symbols. The corresponding barriers at different bias fields are reported in Fig. 5b. For short, one to four unit cell displacements, the barrier height increases with $\Delta x$ from about 0.1 eV for $\Delta x = 1$ up to 0.8 eV for $\Delta x = 4$. The corresponding transition paths closely resemble the continuous displacement of the bubble by a nearby PFM tip shown in Fig. 3a or spontaneous continuous displacements of bubbles triggered by critical thermal fluctuations. For larger distances $\Delta x$, the energy barrier first decays and then sets at a constant value for $\Delta x > 7$ u.c. In this latter case, the NEB transition path corresponds to the bubble teleportation process shown in Fig. 3b. As it can be seen from Fig. 5c, such behavior of the barrier height with the displacement distance is universal for all of the considered bias magnitudes. Furthermore, Fig. 4c clearly shows that the intra-basin barriers decrease with increasing bias magnitude.

Finally, we have also probed the interaction between electric bubbles in the dilute bubble limit at $E_z = 55 \times 10^7$ V/m. For this, we have performed relaxation of the state with two bubbles separated by a distance of up to 20 unit cells. Calculation of the total energies of such states readily yields the pair-wise interaction potential shown in Supplementary Fig. 5. As one can see, this result shows that electric bubbles behave as "hard spheres" - the energy of the system is essentially independent of the distance between the bubbles at large separations and steeply increases when the bubbles are brought close together.

Therefore, the results presented in Fig. 5 confirm the hierarchical energy landscape structure. Moreover, Fig. 5(b) clearly indicates two distinct possibilities for bubble displacements. Namely, at short distances $\Delta x$, our NEB simulations identify a path of a continuous motion similar to the continuous displacements induced by a nearby PFM tip

The situation is drastically different for $\beta < \beta_*$. In this case, the barrier height is sufficiently small for thermal fluctuations to drive the system to a global minimum at each value of $E_z$. The bub-FE transition is thus continuous, as illustrated in Fig. 4d. In other words, at low screening, the energy barriers are low enough for thermal fluctuations to allow for transition between different $N_b$ basins. Furthermore, from Fig. 2e1–e2, we conclude that, at the continuous transition, the thermal

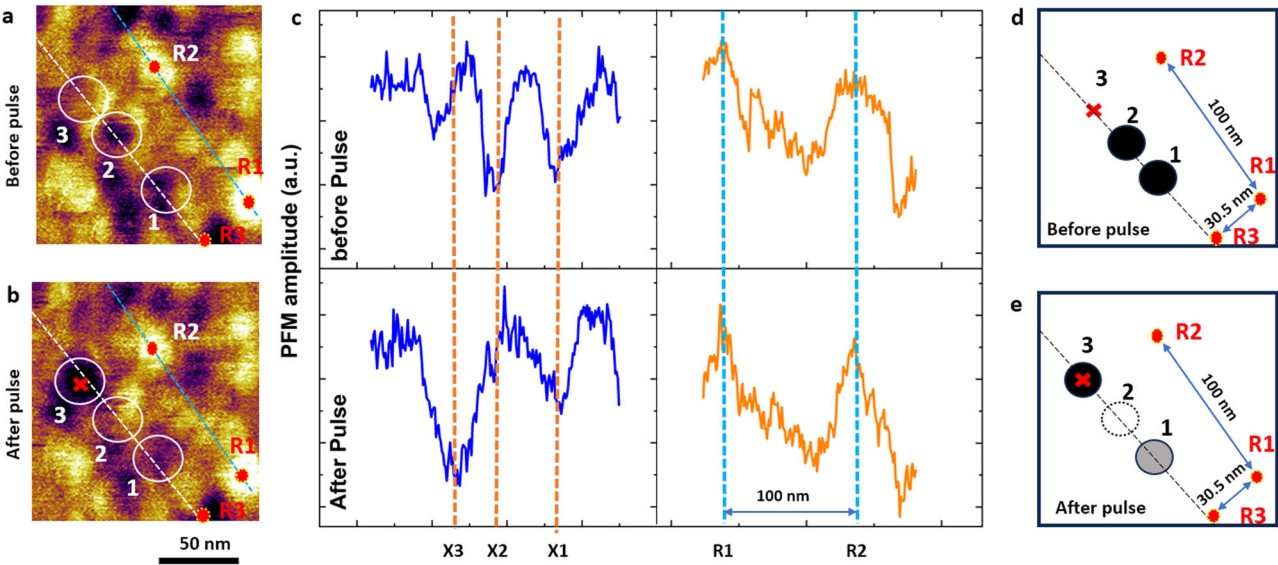

**Fig. 6 | Experiments on bubble teleportation. a** PFM amplitude image of the pristine domain state before applying the PFM pulse. **b** The domain structure after a +2.5 V pulse with a 0.2 s width was applied at the point X1 marked by a red cross. R1–R3 are labeled as reference points. **c** Line profile along points 1–3, and R1 and R2 path before and after applying pulses, are shown in the left and right panel of (**b**), respectively. **d**, **e** Schematic representation of the domain pattern along the line

profile before and after the pulse. The pulse creates a new nanodomain at point X1, while domains at points X2 and X3 either disappear (i.e., point 2) or partly disappear with a faded contrast (i.e., point 3). The location of R1–R3 reference points remains unchanged, with constant distances between each point, suggesting no observable drifting during scanning.

(Fig. 3a). Such continuous motion is characterized by a relatively low energy activation barrier of the order of less than 0.1 eV. Naturally, moving in such a way, the bubble can cover larger distances. For instance, the simulation results shown in Fig. 3a suggest that it would be possible for a moving tip to drag a nearby bubble. Nonetheless, for larger $\Delta x$, our NEB simulations also confirm another possible transition path. Such a path is closely related to the tip-induced teleportation shown in Fig. 3b and consists of the emergence of a bubble at a new location while the original bubble fades out. The mere convergence of the NEB chain to this path confirms that there is a distinct saddle point corresponding to this process. Also, from Fig. 5c, it can be seen that during teleportation, the system overcomes a larger energy barrier. This observation is in line with the larger required magnitude of the tip field (Fig. 3b).

## Discussion

In this study, we have presented evidence for both spontaneous and induced motion of electric bubbles. According to our simulations, the former can be realized under poor screening conditions in the vicinity of the second-order transition from the bubble phase to a homogeneously polarized state. Our simulations predict that, under such conditions, the critical thermal fluctuations push bubbles over the intra-basin energy barriers. Furthermore, we have shown that when screening is enhanced, the greater barrier heights prevent spontaneous bubble dynamics. As a result, the increasing bias triggers a first-order transition between the bubble and homogeneous states. Such transition is also characterized by multiple meta-stable minima corresponding to various static configurations of bubble arrays. Thereby, the vicinity of the first-order transition line can be used to induce bubble motion by, *e.g.*, local electric fields.

Notably, the phenomena that we predict are in line with the previously reported dynamics of magnetic skyrmions and polar dislocations. For instance, the dynamical bubble state at the second-order transition (Fig. 2e1 and Fig. 2e2) closely resembles the liquid phase of magnetic skyrmions[15]. Moreover, our simulations of the tip-induced

continuous bubble dynamics (Fig. 3a) are conceptually similar to the displacements of polar disclinations under the TEM beam[17]. In both cases, the external perturbations distort the local energy landscape, leading to a displacement of the corresponding polar feature. Thereby, we are confident that our predictions will soon find their experimental confirmations in heterostructures comprising ferroelectric (PZT or PTO) and dielectric layers.

For example, in order to experimentally realize a dynamical bubble state, the screening has to be low enough so that an increasing bias field triggers a continuous transition. This can be achieved by varying the thickness of the dielectric spacer. Another possibility lies in varying the height of the intra-basin barriers with an external parameter other than screening. For instance, for thinner ferroelectric layers, the barrier heights will be lower as fewer dipoles will need to be switched during the bubble motion. Finally, another possibility would be to trigger a critical point by increasing temperature rather than increasing the bias. In such a scenario, the height of the energy barriers will not be affected, but the increasing thermal fluctuations could activate bubble displacements.

Another result of our study is the teleportation of electric bubbles, which, thus far, do not have any analogs in ferroelectric or magnetic systems. The origin of this phenomenon lies in the conservation of the number of bubbles that can be achieved only under the right balance of the depolarizing and bias fields. Indeed, bubble teleportation requires that the bubble density in the vicinity of the PFM tip exceeds the equilibrium bubble density. Otherwise, the local field would simply create a new bubble. In other words, external conditions need to assure an energy increase with an increasing number of bubbles (e.g., $E_z > 52 \times 10^7$ V/m in Fig. 5a). An experimental realization of such a scenario can be a challenging task. Nonetheless, in order to prove its feasibility, we have conducted PFM experiments with the epitaxial $Pb(Zr_{0.2}Ti_{0.8})O_3/SrTiO_3/Pb(Zr_{0.2}Ti_{0.8})O_3$ heterostructures deposited on the $La_{0.67}Sr_{0.33}MnO_3$ buffered (001)-oriented stepped $SrTiO_3$ substrate[4]. The thicknesses of the top layer PZT, STO space layer, bottom layer PZT, and LSMO layer are 3 nm, 1 unit cell, 3 nm, and

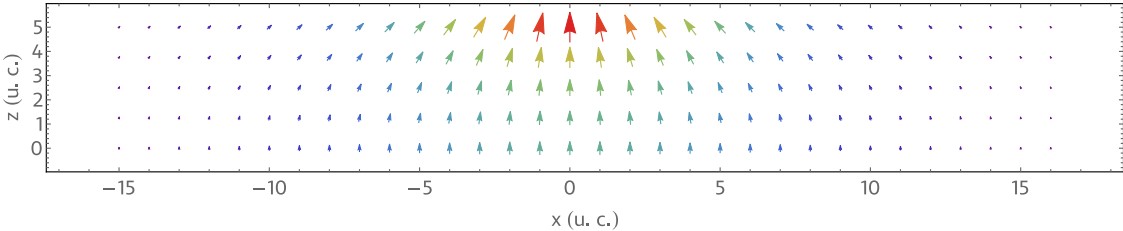

**Fig. 7 | Driving electric field profile.** Arrows indicate the direction of the position-dependent driving electric field within the $y = 0$ plane of the $32 \times 32 \times 5$ supercell for the PFM tip located at $x = y = 0$ and $z = 5.5$ u.c. Colors from purple to red indicate the increasing field magnitude normalized to its maximum value.

15 nm, respectively. To investigate the domain teleportation, a +2.5 V pulse bias with a pulse width of 0.2 s was applied (through bottom electrode to the probe) at a point approximately 10 nm away from the nearest bubble domains (Fig. 6a). The second PFM domain image was captured immediately after applying the pulse bias (Fig. 6b). By comparing the domain configurations before and after applying pulsed bias, we note that the pulse creates a new nanodomain, while one of the nearest nanodomains (e.g., at points $X_2$) disappears. Interestingly, nanodomains located further away from the tip (i.e., at point $X_3$ in Fig. 6) appear suppressed, judging by the partially faded contrast in Fig. 6b. These experiments thereby prove the experimental feasibility of bubble teleportation. Gaining control over this process would be the subject of our further studies.

In summary, in this study, we have presented evidence for particle-like dynamics of polar bubbles. We hope that these findings will inspire new research directions in the domain of polar topologies and open new avenues of technological applications of electric bubbles. For example, given the similarities between bubbles and magnetic skyrmions, one might envision electric analogs of skyrmion-based stochastic computing[20] or racetrack memory[16]. Likewise, we hope that our result will trigger new ideas in the field of magnetic skyrmions.

## Methods

All simulations are performed for a $Pb(Zr_{0.4}Ti_{0.6})O_3$ system with an effective Hamiltonian model described in refs. [1,2,22,29]. The (001) oriented thin-film or slab geometry of ~2 nm (5 u.c.) thickness is mimicked by a $32 \times 32 \times 5$ supercell with periodic boundary conditions imposed along [100] and [010] pseudo-cubic axes. For the results of Fig. 3, the electric boundary conditions along the $z$-axis mimic electrodes that effectively screen 80% of the polarization-induced surface charges. The depolarizing field in each unit cell is computed using an accurate atomistic model[23] that accurately takes into account inhomogeneities of the polarization gradient distribution and hence accounts for intrinsic size effects in low-dimensional ferroelectrics. For all simulations, we assume a compressive strain of −2%. Such value approximately accounts for the mismatch of lattice constants of the cubic phases of strontium titanate (STO) and PZT. A first-principles-based effective Hamiltonian model is used within Monte-Carlo[30] (MC) and molecular dynamics[31] simulations to determine the equilibrium microscopic states and dynamics of local electric dipoles in each perovskite five-atom cell of these supercells. The validity of this approach was demonstrated by previous theoretical studies of ultra-thin PZT films under compressive strains that (1) yield the vortex stripe domains that periodically alternate along [100] (or along [010])[1,2], in agreement with experimental observation[32]; (2) predict a linear dependency between the width of these periodic stripes and the square root of the film's thickness[33], as consistent with measurements[34]; and (3) have also led to the prediction of various topological defects such as vortices[35], dipolar waves[36], bubbles[2] and merons (or convex disclinations)[10,37] in ferroelectrics, that have been experimentally confirmed[4,37,38].

The results presented in Fig. 1b, c and Fig. 2a–c of the manuscript are obtained from Monte Carlo annealing simulations. For each of the considered values of $\beta$, the system is first cooled from 2000 K down to 300 K with 50 K steps under zero external electric field. Then, an external electric field with progressively increasing magnitude is applied. Each run associated with constant values of $\beta$, $T$, and $E$ consists of 40,000 MC sweeps with 20,000 sweeps to be considered as the thermalization period. For both the cooling and the external electric field simulations, the starting configuration for the subsequent parameter value is taken to be the final microscopic state obtained from the preceding run. Fig. 1b, c shows the ground-state dipolar structure at 10 K. In molecular dynamics simulations, we use a predictor–corrector numerical integration scheme[31] with a discrete time step of 0.5 fs.

To demonstrate the field-driven displacements of electric bubbles (Fig. 3), we have used the perturbing potential approximating an electric field generated by a PFM probe. The specific external field model employed here corresponds to the following electric potential[24]

$$\phi = -\frac{2Q}{4\pi\epsilon_0}\left[\frac{1}{1+\epsilon_1/\epsilon_0}\sum_{n=0}^{\infty}\frac{\zeta^n}{\sqrt{(z+2nh)^2+r^2}} - \frac{1-\epsilon_1/\epsilon_2}{(1+\epsilon_1/\epsilon_0)(1+\epsilon_1/\epsilon_2)}\times\sum_{n=0}^{\infty}\frac{\zeta^n}{\sqrt{(2(n+1)h-z)^2+r^2}}\right], \quad (2)$$

$$\zeta = \frac{(1-\epsilon_1/\epsilon_0)(1-\epsilon_1/\epsilon_2)}{(1-\epsilon_1/\epsilon_0)(1-\epsilon_1/\epsilon_2)},$$

where $h$ denotes the film thickness, $\epsilon_0$, $\epsilon_1$, and $\epsilon_2$ denote the vacuum, film, and substrate dielectric permittivities, respectively. The $z$ and $r$ indicate the out-of-plane and radial cylindrical coordinates. This equation is derived under the assumption of the contact mode operation of the tip (the tip located at the surface of the film). The constant $Q$ is determined so as to assure the proper normalization of the corresponding electric field magnitude distribution. Specifically, we require the maximum value of $|\nabla\phi|$ to be equal to the specified magnitude $E$ of the perturbing electric field. An example of the resulting distribution of the local driving electric field is shown in Fig. 7. Note that the total external electric field at each lattice site corresponds to the sum of the bias background field constant within the supercell volume and the position-dependent electric field perturbation.

## Data availability

Data supporting this study is available from the corresponding author on request.

## Code availability

The codes used in this study are available from the corresponding authors on request.

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

## Acknowledgements

S.P., Y.N., L.B., V.G., and N.V. thank the financial support of the DARPA Grant No. HR0011727183-D18AP00010 (TEE Program). S.P., Y.N., and L.B. also thank the Vannevar Bush Faculty Fellowship (VBFF) grant No. N00014-20-1-2834 from the Department of Defense and award No. DMR-1906383 from the National Science Foundation AMASE-i Pro-gram (MonArk NSF Quantum Foundry). The research at the University of New South Wales (UNSW) was supported by an Australian Research Council (ARC) Discovery Project partially supported by the Australian Research Council Centre of Excellence in Future Low-Energy Electronics Technologies (project number CE170100039) and funded by the Australian Government. The presented simulations were made possible thanks to the computational support of the Arkansas High-Performance Computing Center for computational resources.

## Author contributions

S.P., Y.N., and L.B. conceived the study. S.P. and Y.N. designed the simulation strategy, performed the simulations, and analyzed the simulation results. V.G. and Q.Z. developed an experimental metho-dology, carried out the experiments, analyzed the corresponding data, and prepared Fig. 6. N.V. supervised the experiments and contributed to experimental analysis and conceptualization. L.B. supervised the study. S.P. wrote the first draft of the paper. All authors participated in dis-cussions and the writing of the paper.

## Competing interests

The authors declare no competing interests.
