## [Peer Review File · Nature Communications]

Motion and teleportation of polar bubbles in low-dimensional ferroelectricsReviewers' comments:

Reviewer #1 (Remarks to the Author):

In the manuscript titled "Motion and teleportation of bubble skyrmions in low-dimensional ferroelectrics" S. Prokhorenko and coworkers study, using an effective Hamiltonian, the effect of global and local electric field on ferroelectric bubbles. The title and the abstract stress the importance of finding mechanisms to move ferroelectric bubbles which, indeed, is a current, important problem in the field. However, I believe the paper has several problems and does not make a significant contribution towards solving the issue.

1. While the main topic of the paper is the bubble motion, the manuscript is divided in three parts of similar length and the motion is discussed only in the final one. On the other two the authors discuss first the structure of the ferroelectric bubble and in the second they show the transformation of stripe domains into bubbles controlling the screening of the depolarizing field. While the second one is clearly interesting and establishes interesting links to the way skyrmionic phases are detected it is not strongly connected to the motion of the bubbles and the paper could do without it. I must stress that the authors results are interesting but I believe a longer report on that issue showing the phase diagram with field and temperatures would make a nice paper.

2. The authors main results involve the simulation of two kinds of motion that are triggered by placing a piezoresponse force microscopy tip close or far to a ferroelectric bubble. In the first case the bubble moves toward the point of application while in the second the authors report "teleportation" of the bubble.

Neither result is surprising given the well-characterized polar structure of the bubble and the well-understood physics of writing a bubble with a local electric field. The use of such local probes to write cylindrical domains has been already suggested in the literature both by the authors of the manuscript (e.g. their ref. 6) and elsewhere (e.g. Gonçalves et al. *Sci. Adv.* 5, eaau7023, 2019).

The most enticing result, the "teleportation" of the bubble, is directly related to writing a new bubble with a piezoelectric tip which has limited novelty. The simultaneous disappearance of the initial one is only due to carefully balanced electrostatic conditions of the simulation. Neither this nor the more standard motion of the bubble with a close perturbation has nothing to do with the motions of magnetic skyrmions that occur, e.g., in the presence of electric currents.

3. The characterization of the motion is qualitative. The authors discuss its origin with formulas (1) and (2) and mention the electric field threshold to trigger the motion but they do not characterize the displacement characteristics with external factors (how fast it moves, etc.). I believe the paper would benefit from shortening the second part (as mentioned before) and being more quantitative with regards of the movement of the bubble.

In this sense the discussion at the end of page 4 and at the top of page 5 of the condition for teleportation is confusing. For example, they mention the "velocity v of the information propagation in the material" that is described by a characteristic correlation length of dipolar fluctuations and a relaxation time. Could the authors quantify these magnitudes? Is the computational experiment consistent with those independently calculated values?

As a small comment I guess that when the authors mention (top page 5) that "the distance ... separation proportional to $n^{-1/3}$ ", given that the bubbles are placed on a plane they really mean $n^{-1/2}$.

4. The authors use throughout the manuscript the term skyrmion to refer to their ferroelectric bubbles and this is inconsistent with previous literature. While the authors prove that their bubbles have well-characterized plane-by-plane topological charge like in Ref. 5, in that paper it is also argued that this is not enough to call your system a skyrmion as this would not be consistent with the terminology used in magnetic skyrmions. Essentially a magnetic skyrmion is different from a chiral bubble in that skyrmions produce chiral macroscopic phases. Given that the bubbles reported by the authors are not chiral my suggestion is to refrain from using the term skyrmion and use topological bubble.

Reviewer #2 (Remarks to the Author):

This manuscript proposed to Nature Communications journal by S. Prokhorenko with all respected coauthors brings to the reader's attention an interesting subject of motion, formation and annihilation of small ferroelectric nanodomains. The paper is based on computational simulations with an established, effective Hamiltonian approach. More exactly, authors have explored behavior of ultrathin film of PZT under conditions favoring formations of so-called bubble nanodomains, following the original discovery made earlier in the same group (Ref. 2 and 3). By the level of presentation, motivation, insights in the phenomena observed, and their explanation to the readers, it certainly belongs to the top 10 percent of the best papers based on the effective Hamiltonian simulations in ferroelectrics, and I would warmly recommend acceptance of the paper.

There are, however, several issues that are desirable to be improved, for the sake of the better understanding of all of us and also for a better recognition of the work done by authors, even though the group surely involve the world's best experts in the field.

In particular:

1. The abstract and the first paragraph is referring to these bubble domains (term used in earlier Refs.2,3,4,6) subsequently as - polar bubble domains, -electric bubble skyrmions, -bubble skyrmions, -skyrmion domains, - polar skyrmions, -ferroelectric bubble domains, -polar bubbles, -electric skyrmions, -bubble domain skyrmions, and simply -bubble domains; later in the text also as -skyrmion bubble domains. I did not find a clear definitions of these terms so I expect that these 11 terms simply refer to the bubble domains subject to the present simulation, therefore, I wonder if this richness is really appropriate here, in particular, in the situation when we have several qualitatively different type of similar objects in the field already and some of these terms have been already used (perhaps also too vaguely) for a very different type of nanodomains (Ref. 5 by Das, for example). Thus: why not to stick to the formerly given bubble nanodomain (Ref. 2) not to make a more confusion and difficulty in communicating the differences?

2. The present paper is novel by ascribing Skyrmion charge to the <<bubble domain of Ref.2>>. This is useful and valuable to be pointed out, I think. I feel that authors could draw a more attention to this fact but in any case so far there is no clear definition of this generalized charge nor it is explained how is exactly calculated. Authors are referring to Ref. 5 which uses the usual definition, exploited to characterize magnetic magnetic skyrmions or

ferroelectric bubbles of Ref. 5, and this standard definition relies on the 2D integration of the Pontryagin density over a real space plane crossing the axis of the skyrmion. The very same definition, however, does not apply to the present case – all polarization directions within a plane perpendicular to the axis of the present bubble do not cover the full solid angle and thus the standard definition yields zero. Thus, according to the established definition (Nagaosa and Tokura 2013 paper, for example), the present bubbles do not have the same skyrmion number as those of Ref. 1 or 5. Ignoring this difference could be misleading, and it is a pity not to mention new definition of authors.

3. Figure 3 should be better associated with temporal profiles of BOTH the average polarization and averaged Skyrmionic charge (3D definition probably specified in the revised paper by authors) for BOTH the bottom and top series of processes (drift vs teleportation). This can replace the Figure 5. I missed an explanation about the variation of the Skyrmion charge, fluctuations of the polarization may not be so surprising in the end as the variations of the Skyrmion charge, as the latter quantity is often argued to be an integer.

4. Teleportation is a nice term for the title of the paper, I definitely agree to keep it, but more justification is needed in the text. Why the reader should consider that formation of a new bubble should be interpreted as a teleportation of the same bubble? What if one bubble was simply formed by the human-made tailored perturbation somehow by chance following the stimulated decay of the latter? How do we know that the former and latter bubble are related if the overall skyrmion charge is violated? How much it is violated? It climbs from 1 to 2 and back to 1? If there would be 2 equally distant bubbles from the tip in the beginning instead of only 1, will only one of them annihilate?

Reviewer #3 (Remarks to the Author):

This manuscript describes a series of atomistic simulations with primary focus on understanding the electric field response of bubble domains in PZT. The manuscript notes the observation of both continuous motion and discontinuous creation/annihilation. Implications for devices are scattered throughout.

To begin, this reviewer agrees that such features are intrinsically interesting and important and studies of the dynamics of them are both timely and important. This immediately places the content of this paper in a realm of interest to the broader community. This said, however, the reviewer would also caution against the use of language like “teleportation” and overselling such ideas. Furthermore, there are really just 2 contributions of this work – Figures 2 and 3 (as Figure 1 is close to work of the authors from prior papers) and the paper takes its time getting to the new results. In turn, additional details of the physical mechanisms at play in Figure 3 seem warranted and honest assessment of the universality of the effects (and what role the simulation setup plays in this) seems necessary. Additional comments are provided below:

- There is a rather long description (page 2, left column) of all the features in Figure 1 that one needs to notice. Despite a great interest in these points, this reader found it hard to

clearly see and mesh those descriptions with the pictures in Figure 1. Perhaps both better calls to specific figures and zoom-ins that highlight the features are required to better illustrate these points to the average reader.

- The polar structures in Figure 1 do not seem to match in make-up the polar skyrmion-like structures reported recently. It is not clear that the bubble domain in Figure 1a would have a non-zero skyrmion number. Perhaps some words on the differences of these bubble domains versus the reported skyrmion-like structures are required.

- Figure 2 – First, the caption suggests a “(c)” panel, but there is no (c) panel labeled. Second, perhaps labeling the evolution from left-to-right in the two rows in b with the variable that is changing would be helpful. Third, what are the length scales of the images in b? How are these images produced from the simulation which is 3D?

- While the findings of Figure 2 are interesting, practical connection to experiment is lost on the reader. Can the authors provide contextualization for how changing beta in the manuscript can be accomplished in an experiment?

- Figure 3 – Why was a PFM-like field profile chosen? Do you think this could be responsible for some of the observations? Would the same things happen in uniform out-of-plane fields? Clearly the gradient in field is key, but can temporally varying fields achieve the same thing?

- Figure 3b – First, I think the use of teleportation is likely unfounded and potentially misleading or confusing to the community. I much prefer the more reasonable description of creation-annihilation used in the text. Instead of using such a controversial word, it would seem better to merely explain the physics of this observation – which as it currently stands really doesn't provide much insight. Second, what is the effect of the simulation cell size on this observation? Of the boundary conditions used? Is this observation always valid or only in a few instances of special conditions?

Response to the report of Reviewer #1:

In the manuscript titled "Motion and teleportation of bubble skyrmions in low-dimensional ferroelectrics" S. Prokhorenko and coworkers study, using an effective Hamiltonian, the effect of global and local electric field on ferroelectric bubbles. The title and the abstract stress the importance of finding mechanisms to move ferroelectric bubbles which, indeed, is a current, important problem in the field. However, I believe the paper has several problems and does not make a significant contribution towards solving the issue.

Response:

We thank Reviewer 1 for careful reading and assessment of our manuscript. We absolutely agree with Reviewer 1 in that motion of electric bubbles is a very important topic. We appreciate the constructive feedback provided by Reviewer 1 which leads us to a major revision of the manuscript.

We hope that our answers and revisions will convince Reviewer 1 that our revised study makes a significant step in understanding the motion of polar bubbles.

Comment 1.

While the main topic of the paper is the bubble motion, the manuscript is divided in three parts of similar length and the motion is discussed only in the final one. On the other two the authors discuss first the structure of the ferroelectric bubble and in the second they show the transformation of stripe domains into bubbles controlling the screening of the depolarizing field. While the second one is clearly interesting and establishes interesting links to the way skyrmionic phases are detected it is not strongly connected to the motion of the bubbles and the paper could do without it. I must stress that the authors results are interesting but I believe a longer report on that issue showing the phase diagram with field and temperatures would make a nice paper.

Response:

We thank Reviewer 1 for pointing out the weak relation between the second part of our paper (phase diagram results) and the main topic of the study, i.e., bubble motion.

To address the issues pointed out by Reviewer 1, we have:

- 1.) re-written the introduction making the discussion of the bubble structure as succinct as possible, while, at the same time extending the introductory discussion of the bubble motion vis-à-vis the reported motion of other topological structures.

- 2.) Regarding the phase diagram part, we have followed the Reviewer's suggestion to extend the analysis. While further exploring the phase diagram results, we have found one previously overlooked feature that we find of utmost importance for the bubble motion. Such feature is the tri-critical point on the **bub-FE** line. As we show in the revised manuscript, this tri-critical point separates the regime of spontaneous bubble motion and the regime where bubbles can be displaced by external perturbations.

In the revised manuscript we made sure to focus the phase diagram discussion on the features directly related to the bubble motion. This discussion is meant to show that:

- a. bubble motion can be realized at the **bub-FE** line where the bubble density is low
- b. depending on the screening conditions, such motion can be spontaneous (driven by thermal fluctuations) or induced by an external stimuli. The cross-over occurs at the tri-critical point.

We would also like to thank Reviewer 1 for judging that our phase diagram results are "interesting" and encouraging us to make a longer report on this subject. We have followed the suggestion of Reviewer 1 and are currently collecting data for such a report.

Comment 2.

The authors main results involve the simulation of two kinds of motion that are triggered by placing a piezoresponse force microscopy tip close or far to a ferroelectric bubble. In the first case the bubble moves toward the point of application while in the second the authors report "teleportation" of the bubble.

Neither result is surprising given the well-characterized polar structure of the bubble and the well-understood physics of writing a bubble with a local electric field. The use of such local probes to write cylindrical domains has been already suggested in the literature both by the authors of the manuscript (e.g. their ref. 6) and elsewhere (e.g. Gonçalves et al. Sci. Adv. 5, eaau7023, 2019).

The most enticing result, the "teleportation" of the bubble, is directly related to writing a new bubble with a piezoelectric tip which has limited novelty. The simultaneous disappearance of the initial one is only due to carefully balanced electrostatic conditions of the simulation. Neither this nor the more standard motion of the bubble with a close perturbation has nothing to do with the motions of magnetic skyrmions that occur, e.g., in the presence of electric currents.

Response:

We absolutely agree with Reviewer 1 in that writing of a bubble domain with local electric field is an expected result. However, in our work we discuss displacement, i.e., *motion*, of already existing bubbles. In our opinion, this is a different and previously unreported process. In our work we report conditions for which a local field does not write new bubbles but rather displaces them.

Regarding the bubble teleportation, we appreciate that Reviewer 1 finds this result reasonable given the considered electrostatic conditions. To show that such conditions can be reached in practice, we have added experimental demonstration of such process.

In fact, we absolutely agree that teleportation makes a lot of sense. At the same time, we would like to note that this result is nonetheless very surprising since we are not aware of any reported analogue of this process neither in magnetic nor in other topological soliton systems.

Finally, the reported motion of bubbles is indeed different from the current driven motion of magnetic skyrmions. Because of it, in the revised manuscript we do not claim that our results can be used to construct direct electric analogues of skyrmion racetrack memories. However, we do not find that this limits the novelty of our study since our results open avenues for other applications such as bubble-based reconfigurable electronic circuits.

Finally, an additional novel result is now reported in our revised work – the discovery of a spontaneous bubble motion giving rise to a bubble liquid state. Therefore, the revised manuscript describes:

- the thermal or fluctuation-induced bubble motion
- the electric-field induced bubble motion
- how to achieve the cross-over between these two types of dynamics at room temperature

To the best of our knowledge, all of these results are novel, and we believe that they make a significant advancement in the important topic of the electric bubble motion.

Comment 3.

The characterization of the motion is qualitative. The authors discuss its origin with formulas (1) and (2) and mention the electric field threshold to trigger the motion but they do not characterize the displacement characteristics with external factors (how fast it moves, etc.). I believe the paper would benefit from shortening the second part (as mentioned before) and being more quantitative with regards of the movement of the bubble.

In this sense the discussion at the end of page 4 and at the top of page 5 of the condition for teleportation is confusing. For example, they mention the "velocity v of the information propagation in the material" that is described by a characteristic correlation length of dipolar fluctuations and a relaxation time. Could the authors quantify these magnitudes? Is the computational experiment consistent with those independently calculated values?

As a small comment I guess that when the authors mention (top page 5) that "the distance ... separation proportional to $n^{-1/3}$ ", given that the bubbles are placed on a plane they really mean $n^{-1/2}$.

Response:

We thank Reviewer 1 for bringing our attention to this issue. To address this critique, we have removed the qualitative formulas (1) and (2) and opted to discuss the physics of the different types of bubble motion from the perspective of the energy landscape structure. Specifically, the numerical characterization of the energy landscape and its relation to bubble motion is now described in the section “Origins of tri-critical behavior and underlying bubble dynamics”.

Moreover, in the revised version we provide the computed velocity of bubble displacement during the continuous motion process. The graph below shows the calculated evolution of the change in the position X of the bubble center with time during the continuous displacement process as shown in Fig. 3a for the tip field magnitude of $26 \times 10^7 \text{V/m}$.

Collecting such data over 50 independent runs and calculating the bubble velocity from the linear fits (e.g., red line in the figure above) yields an average bubble velocity of 2.040 nm/ps or 2040 m/s.

To further characterize the continuous motion of bubbles, we have performed the simulations described above for a range of tip field magnitudes ranging from $20 \times 10^7 \text{V/m}$ to $30 \times 10^7 \text{V/m}$. The resulting dependence of bubble velocity on the tip field magnitude is shown in the figure below. For each driving field magnitude, 20 runs were performed to estimate the velocity.

Comment 4.

The authors use throughout the manuscript the term skyrmion to refer to their ferroelectric bubbles and this is inconsistent with previous literature. While the authors prove that their bubbles have well-characterized plane-by-plane topological charge like in Ref. 5, in that paper it is also argued that this is not enough to call your system a skyrmion as this would not be consistent with the terminology used in magnetic skyrmions. Essentially a magnetic skyrmion is different from a chiral bubble in that skyrmions produce chiral macroscopic phases. Given that the bubbles reported by the authors are not chiral my suggestion is to refrain from using the term skyrmion and use topological bubble.

Response:

This is a very pertinent and wise remark. We have followed the suggestion of the Reviewer 1 and refrained from using the term “polar skyrmion” in the revised manuscript. We also address the interested reader to our recent reviews [Nat. Mater. **22**, 553 (2023) and Rev. Mod. Phys. **95**, 025001 (2023)] where these terminological issues are discussed.

Response to the report of Reviewer #2:

This manuscript proposed to Nature Communications journal by S. Prokhorenko with all respected co-authors brings to the reader's attention an interesting subject of motion, formation and annihilation of small ferroelectric nanodomains. The paper is based on computational simulations with an established, effective Hamiltonian approach. More exactly, authors have explored behavior of ultrathin film of PZT under conditions favoring formations of so-called bubble nanodomains, following the original discovery made earlier in the same group (Ref. 2 and 3). By the level of presentation, motivation, insights in the phenomena observed, and their explanation to the readers, it certainly belongs to the top 10 percent of the best papers based on the effective Hamiltonian simulations in ferroelectrics, and I would warmly recommend acceptance of the paper.

There are, however, several issues that are desirable to be improved, for the sake of the better understanding of all of us and also for a better recognition of the work done by authors, even though the group surely involve the world's best experts in the field.

Response:

We thank Reviewer 2 for the careful reading of the manuscript and the constructive criticism of our work. We appreciate his/her positive assessment of our study and the recommendation to accept this paper.

To address the concerns of the 3 Reviewers, we made a major revision of the manuscript. We have added new results and have re-written multiple parts of the paper. At the same time, we would like to assure Reviewer 2 that all of the results described in the previous version are still present in the revised manuscript. We strongly believe that, thanks to the constructive criticism of the Reviewers, the quality of the manuscript has significantly improved while preserving the strong points of the previous version.

Comment 1.

The abstract and the first paragraph is referring to these bubble domains (term used in earlier Refs.2,3,4,6) subsequently as
- polar bubble domains, -electric bubble skyrmions, -bubble skyrmions, -skyrmion domains, -polar skyrmions, -ferroelectric bubble domains, -polar bubbles, -electric skyrmions, -bubble domain skyrmions, and simply -bubble domains; later in the text also as -skyrmion bubble domains. I did not find a clear definitions of these terms so I expect that these 11 terms simply refer to the bubble domains subject to the present simulation, therefore, I wonder if this richness is really appropriate here, in particular, in the situation when we have several qualitatively different type of similar objects in the field already and some of these terms have been already

used (perhaps also too vaguely) for a very different type of nanodomains (Ref. 5 by Das, for example). Thus: why not to stick to the formerly given bubble nanodomain (Ref. 2) not to make a more confusion and difficulty in communicating the differences?

Response:

We thank Reviewer 1 for bringing this very wise remark to our attention. There is indeed a terminological issue which has been addressed in recent reviews [Nat. Mater. **22**, 553 (2023) and Rev. Mod. Phys. **95**, 025001 (2023)].

Given that the issue was already discussed in the literature, we decided not to include the corresponding discussion in this work and rather redirect interested readers to the relevant papers. This said, we absolutely agree with Reviewer 1 and have used the term bubble throughout the paper.

Comment 2.

The present paper is novel by ascribing Skyrmion charge to the $\langle \rangle$. This is useful and valuable to be pointed out, I think. I feel that authors could draw a more attention to this fact but in any case so far there is no clear definition of this generalized charge nor it is explained how is exactly calculated. Authors are referring to Ref. 5 which uses the usual definition, exploited to characterize magnetic magnetic skyrmions or ferroelectric bubbles of Ref. 5, and this standard definition relies on the 2D integration of the Pontryagin density over a real space plane crossing the axis of the skyrmion. The very same definition, however, does not apply to the present case – all polarization directions within a plane perpendicular to the axis of the present bubble do not cover the full solid angle and thus the standard definition yields zero. Thus, according to the established definition (Nagaosa and Tokura 2013 paper, for example), the present bubbles do not have the same skyrmion number as those of Ref. 1 or 5. Ignoring this difference could be misleading, and it is a pity not to mention new definition of authors.

Response:

We thank Reviewer 2 for pointing out this issue. In fact, similarly to the methodological “pickle” mentioned in Comment 1, the calculation of the Skyrmion number for bubbles has been discussed in Nat. Mater. **22**, 553 (2023) and Rev. Mod. Phys. **95**, 025001 (2023).

To address this concern, we refer an interested reader to these papers in the Introduction of our manuscript.

Comment 3.

Figure 3 should be better associated with temporal profiles of BOTH the average polarization and averaged Skyrmionic charge (3D definition probably specified in the revised paper by authors) for BOTH the bottom and top series of processes (drift vs teleportation). This can replace the Figure 5. I missed an explanation about the variation of the Skyrmion charge,

fluctuations of the polarization may not be so surprising in the end as the variations of the Skyrmion charge, as the latter quantity is often argued to be an integer.

Response:

We thank the Reviewer for bringing our attention to this oversight. We have now included the requested graphs in the new Figure 3 of the manuscript. For convenience, we also provide the calculated dependence of the Skyrmion number (N_{Sk}) and the normalized out-of-plane component of polarization (P_z/P_0) below

In this figure, panels (c) and (d) correspond to the continuous motion and teleportation processes, respectively. Regarding the Skyrmion number, we further provide the corresponding values within the $z=2$ and $z=4$ planes of the supercell. This allows to reveal that the switching during the teleportation process is inhomogeneous – the nucleation of the bubble beneath the tip starts at the top interface where the driving field is more intense.

Comment 4.

Teleportation is a nice term for the title of the paper, I definitely agree to keep it, but more justification is needed in the text. Why the reader should consider that formation of a new bubble should be interpreted as a teleportation of the same bubble? What if one bubble was simply formed by the human-made taylored perturbation somehow by chance following the stimulated decay of the latter? How do we know that the former and latter bubble are related if the overall skyrmion charge is violated? How much it is violated? It climbs from 1 to 2 and back to 1? If there would be 2 equally distant bubbles from the tip in the beginning instead of only 1, will only one of them annihilate?

Response:

This is a very interesting question. In fact, since all bubbles are identical, there is no way to distinguish between the transfer of the original bubble and a synchronous destruction/re-creation of a new bubble.

Nonetheless, we found a way to show that teleportation process is distinct from the creation/erasure process. For this, we have performed nudged elastic band (NEB) simulations that allow to probe different transition paths between the two states. In this method, no external perturbation is applied to the system.

The two end states of the path (states A and B) that we considered corresponded to a single bubble located at different points of the supercell as schematically shown below:

Running multiple simulations with different random seeds, we were able to identify three distinct paths connecting states A and B. These correspond to:

- Path 1: creation of a bubble, followed by annihilation of the initial bubble
- Path 2: simultaneous creation/annihilation process that we call teleportation
- Path 3: annihilation of the initial bubble followed by a creation of the new bubble

Such processes are schematically represented in the diagram below where C A and T stand for creation, annihilation and teleportation of a bubble:

The energy diagram along such paths obtained from NEB simulations is shown below:

Where ξ denotes the normalized distance from the initial state (reaction coordinate) along the converged path.

The fact that the NEB algorithm was able to converge to a teleportation path means that there is a distinct saddle point connecting single-bubble states A and B. To reach this point, one needs to apply a large-magnitude field perturbation as described in our manuscript. In our opinion, this proves that teleportation is a process distinct from creation followed by annihilation (or vice versa) of a bubble.

Coming back to the question of whether our use of the term “teleportation” is justified, we would like to give two arguments. Firstly, in our opinion, such name for this distinct process would be appropriate given the common intuition about teleportation. After all, teleportation means that an identical object appears somewhere else independently of whether it was re-created or transferred. Secondly, there is already a precedent of this term being employed in physics – the so-called “quantum teleportation” phenomenon where a state of a quantum particle (photon) is recreated in a faraway location. In the latter case, the quantum particle is not transferred, but rather an identical particle (or state) is re-created. This process is akin to bubble teleportation reported in our work.

Response to the report of Reviewer #3:

This manuscript describes a series of atomistic simulations with primary focus on understanding the electric field response of bubble domains in PZT. The manuscript notes the observation of both continuous motion and discontinuous creation/annihilation. Implications for devices are scattered throughout.

To begin, this reviewer agrees that such features are intrinsically interesting and important and studies of the dynamics of them are both timely and important. This immediately places the content of this paper in a realm of interest to the broader community. This said, however, the reviewer would also caution against the use of language like “teleportation” and overselling such ideas.

Response:

We thank Reviewer 3 for carefully reading our manuscript and providing constructive criticism. We also thank Reviewer 3 for highlighting the importance of the topic and its interest to the broad readership of Nature Communications.

At this point we would like to note that, based on the feedback of all Reviewers, we have made a major revision of our manuscript. While we discuss all results from the previous version, we have also added new discussions that add novelty to our work. We hope that these extensive revisions will convince Reviewer 3 that our work can be published as a research article in Nature Communications.

We also appreciate the concern of Reviewer 3 about the use of the term teleportation. Initially, we have used such term in analogy with the previously reported “quantum teleportation” which can be described as re-creation of a quantum particle’s state in a faraway location. In our opinion, the synchronous bubble creation/annihilation process that we report in our work is akin to this process. Moreover, in our opinion the common understanding of the term teleportation fits rather well such a distinct transition path between the two single-bubble states. We provide further arguments in favor of using this term in our answer to comment 4 of Reviewer 2.

We would also like to assure Reviewer 3 that we are not at all trying to oversell our ideas. Instead, our motivation is to convey our ideas in the simplest way possible so that the main results can be quickly grasped by a broad audience. Also, we find that just a touch of informality in research articles (as it used to be in the days of Dirac, Feynman, Landau, Einstein and other great scientists) brings an additional spark to physics which ignites the imagination of younger students to pursue research carriers.

This said, we also fully understand the opposite point of view and are not against removing the term teleportation if it does not comply with the policy of the journal.

Comment 1:

Furthermore, there are really just 2 contributions of this work – Figures 2 and 3 (as Figure 1 is close to work of the authors from prior papers) and the paper takes it time getting to the new results. In turn, additional details of the physical mechanisms at play in Figure 3 seem warranted and honest assessment of the universality of the effects (and what role the simulation setup plays in this) seems necessary. Thereby, 5 out of 6 figures contain exclusively novel results. Figure 1 serves as an introduction and does not contain any new concepts or data.

Response:

We thank Reviewer 3 for this comment. In the revised version, we have re-written the introduction and the phase diagram discussion to make a faster transition to the main contributions of this study.

Particularly, the revised introduction contains an original discussion pertaining to the motion of electric bubbles vis-à-vis the motion of magnetic skyrmions and polar dislocations. The discussion of new results starts on page 2.

Figure 2 and the accompanying discussion has been also modified. Here, we have added the discussion of the dynamical bubble density hysteresis and the structural snapshots of bubble liquid-like state. Similarly, we have added new Figs. 4 and 5 describing and characterizing the physical mechanisms at play in the new Figs. 2 and 3. Additionally, experimental results have been added in the new Fig. 6.

Finally, several new and important results have been added. Particularly, we describe the discovery of a liquid-like state wherein bubble move spontaneously. This state is linked to our discussion of perturbation-induced motion of bubbles via the phase diagram (Fig. 2) and the energy landscape mappings (Figs. 4 and 5). Apart from explaining the described phenomena, the presented energy mappings also provide a new viewpoint on polar topologies which, as we hope, will prove to be useful in the field.

Additional comments:

- 1) There is a rather long description (page 2, left column) of all the features in Figure 1 that one needs to notice. Despite a great interest in these points, this reader found it hard to clearly see and mesh those descriptions with the pictures in Figure 1. Perhaps both better calls to specific figures and zoom-ins that highlight the features are required to better illustrate these points to the average reader.

Response:

We fully agree with Reviewer 3. To address this concern, we have reduced the discussion of

Figure 1. Instead, we redirect the interested reader to recent reviews discussing the structure and topology of polar bubbles.

- 2) The polar structures in Figure 1 do not seem to match in make-up the polar skyrmion-like structures reported recently. It is not clear that the bubble domain in Figure 1a would have a non-zero skyrmion number. Perhaps some words on the differences of these bubble domains versus the reported skyrmion-like structures are required.

Response:

To address this concern we have added a short mention of the two most common skyrmion-like structures in the introduction. Also, in order to reduce the introduction to a minimum, we have included references to articles describing in great detail the topology and structure of electric and skyrmion bubbles in ferroelectric materials.

- 3) Figure 2 – First, the caption suggests a “(c)” panel, but there is no (c) panel labeled. Second, perhaps labeling the evolution from left-to-right in the two rows in b with the variable that is changing would be helpful. Third, what are the length scales of the images in b? How are these images produced from the simulation which is 3D?

Response:

We thank Reviewer 3 for these suggestions and for pointing out the inconsistency in panel labels. To address these concerns, we have made the corresponding revisions in the layout and the caption of Figure 2.

- 4) While the findings of Figure 2 are interesting, practical connection to experiment is lost on the reader. Can the authors provide contextualization for how changing beta in the manuscript can be accomplished in an experiment?

Response:

This is indeed a very important point. Experimentally, the screening conditions can be changed by introducing overlayers of different thickness on the top and/or bottom interfaces of the PZT film. A typical material used for such overlayers is SrTiO₃. This is now indicated on page 2 of the manuscript.

- 5) Figure 3 – Why was a PFM-like field profile chosen? Do you think this could be responsible for some of the observations? Would the same things happen in uniform out-of-plane fields? Clearly the gradient in field is key, but can temporally varying fields achieve the same thing?

Response:

We have chosen a PFM-like profile because this is the most experimentally feasible option for creating a non-uniform electric field. Regarding a uniform out-of-plane field, we now show that the spontaneous bubble dynamics can be induced in such case under poor

screening conditions. Finally, the question regarding a temporally varying field is very interesting. We believe that such fields will certainly lead to non-trivial dynamics. However, this question lies out of the scope of this work.

- 6) Figure 3b – First, I think the use of teleportation is likely unfounded and potentially misleading or confusing to the community. I much prefer the more reasonable description of creation-annihilation used in the text. Instead of using such a controversial word, it would seem better to merely explain the physics of this observation – which as it currently stands really doesn't provide much insight. Second, what is the effect of the simulation cell size on this observation? Of the boundary conditions used? Is this observation always valid or only in a few instances of special conditions?

Response:

We thank Reviewer 3 for these important comments. We have outlined our position regarding the “teleportation” term in the beginning of our response to Reviewer 3 and in our answer to comment 4 of Reviewer 2. We would not mind at all removing this term if it does not align with the journal policy.

At the same time, we have performed extensive nudged elastic band simulation showing that the phenomenon in question is distinct from the two-step creation/annihilation process. Also, we have provided our view on the physical explanation in the new section of the manuscript named “Origin of tri-critical behavior and underlying bubble dynamics”.

The conditions under which the creation-annihilation can be observed are discussed in the last section:

- (1) The screening needs to be sufficiently high to ensure first-order character of the bubble-monodomain transition (Fig. 2)
- (2) The system has to be found in a meta-stable state with a higher-than-equilibrium density of bubbles (Fig. 4c)
- (3) The magnitude of the PFM pulse has to be high enough to exceed the energy barrier associated with the teleportation process (Fig. 5).

To probe the effects of the supercell size, we have performed the simulations presented in Fig. 3 for a 64x64x5 supercell and did not find any qualitative differences with 32x32x5 supercell simulations. This observation is consistent with the investigation of finite-size effects discussed in Refs. 13 and 10 of the manuscript.

REVIEWERS' COMMENTS

Reviewer #1 (Remarks to the Author):

In their response letter and revised manuscript the authors have implemented a rather dramatic change in the text. Contrary to the previous version I believe there is now a strong focus on the movement of the skyrmions and the computational results predicting the conditions and velocity of the movement have been complemented with experimental results showing the creation/annihilation of one of the bubbles with a PFM tip.

Another important improvement has been clearly connecting the second part of the manuscript to the movement of the bubbles in the third part. The latter now quantifies the movement. While I still believe that the presented results do not solve the problem of polar bubble movement they certainly characterize aspects of the bubbles dynamic behavior that are new and deserve attention. I also believe that the authors made significant efforts to demonstrate that, under some conditions, the disappearance of the old bubble/appearance of a new one is synchronized to a very high extent. Thus, I recommend publication.

Reviewer #2 (Remarks to the Author):

The think that the authors have addressed sufficiently all the suggestions and criticism and the revised manuscript can accepted for publication